# Survey of Natural Enemies of the Invasive Boxwood Moth *Cydalima perspectalis* in Southwestern Mediterranean Europe and Biocontrol Potential of a Native *Beauveria bassiana* (Bals.-Criv.) Vuill. Strain

**DOI:** 10.3390/insects13090781

**Published:** 2022-08-29

**Authors:** Carmen López, Sandra Las Heras, Inmaculada Garrido-Jurado, Enrique Quesada-Moraga, Matilde Eizaguirre

**Affiliations:** 1Department of Crop and Forest Sciences, University of Lleida-Agrotenio Center, Av. Al. Rovira Roure 191, 25198 Lleida, Spain; 2Orius Soluciones Entomológicas, 17800 Olot, Spain; 3Department of Agronomy, University of Cordoba, Campus de Rabanales Building C4 “Celestino Mutis”, 14071 Cordoba, Spain

**Keywords:** boxwood moth, native natural enemies, *Beauveria bassiana*, *Compsilura concinnata*, park, garden, treatment

## Abstract

**Simple Summary:**

*Cydalima perspectalis* (Lepidoptera: Crambidae), a species native to East Asia that feeds exclusively on *Buxus* spp., was first detected in Southwestern Germany and the Netherlands in 2007. Today, it is present in many countries in Europe. Since its detection, numerous researchers have studied the biology and the present or potential number of generations of *C. perspectalis* in European countries. However, less research has been devoted to detecting possible native biotic factors of mortality (predators, parasitoids, entomopathogenic microorganisms or nematodes) capable of reducing the pest populations. In this work, indigenous natural enemies in the boxwood of the southwest Mediterranean region were collected and identified. One of them, a native *Beauveria bassiana* strain from infected *C. perspectalis* larvae, showed potential for controlling this invasive crambid.

**Abstract:**

*Cydalima perspectalis* (Lepidoptera: Crambidae), a species native to East Asia, has been especially devastating in the Mediterranean region and Catalonia, northeast Spain, where *Buxus sempervirens* is an essential component of the natural forest. As an invasive species, the lack of biotic mortality factors in the arrival region has been one of the main factors allowing its expansion. Therefore, this study aimed to collect and identify possible indigenous natural enemies adapting to the new species in the boxwood of the southwest Mediterranean region. Later, the efficacy of some of the collected species for controlling *C. perspectalis* larvae was tested in laboratory conditions. The larval collection was carried out in successive years in the boxwood of the region. Several collected larvae were infected with an entomopathogen, *Beauveria bassiana*, or parasitized by *Compsilura concinnata*, both common in native Lepidoptera caterpillars. The *B. bassiana* strain was found to be highly virulent against the developed larvae of *C. perspectalis*, which suggests that *B. bassiana* may be an effective treatment in parks and gardens when the first overwintering larvae are detected. The biology of the parasitoid identified is not very well known in Europe, which suggests the necessity of studying its biology and alternative hosts in the region in order to improve its population.

## 1. Introduction

The boxwood moth, *Cydalima perspectalis* (Walker, 1859) (Lepidoptera: Crambidae), a species native to East Asia that feeds exclusively on *Buxus* spp., was first detected in southwestern Germany and the Netherlands in 2007 [1]. The species arrived in Germany on ornamental box plants, and from there, it spread to nearly all European countries, affecting ornamental plants but also natural boxwood forests in Central Europe and in the Mediterranean region [2], where the common boxwood, *Buxus sempervirens* L., is an essential component of the forest. In Catalonia, northeast Spain, the invasive species has been especially devastating [3].

Among the reasons that explain the expansion of an invasive species in regions far from its origin are the possibility of its dispersal by world trade, the presence of host plants in the destination region that facilitate its implantation, favorable climatic conditions, and the lack of biotic mortality factors in the region of arrival have often been mentioned [4]. The invasion of Europe by *C. perspectalis* has been made possible by the occurrence of all of the factors described above, favoring the implantation and expansion of the species. Very often, when an exotic species arrives in a new region, few or no indigenous natural enemies are capable of feeding on the invader [5]. However, the existing natural enemies can later adapt to the exotic species, which may present an alternative potential food source for them [6]. In agricultural systems, there are several examples that confirm this last statement. This is the case of indigenous natural enemies used against invasive species in the indoor and outdoor vegetal crops that are often exposed to exotic pests [7] or the host-mediated spread of the entomopathogenic fungus *Beauveria bassiana* (Bals.-Criv.) Vuill., 1912 (Ascomycota: Hypocreales) by an invasive palm pest [8]. Several studies have also been carried out to understand how indigenous natural enemies adapted to invasive species such as *Operophthera brumata* (L., 1758) in the USA [9] or recently *Cameraria ohridella* (Deschka and Dimić, 1986) in Italy [10]. A natural response to the arrival of an invasive pest has been described with respect to *Tuta absoluta* (Meyrick, 1917) (Lep. Gelechiidae), which became a devastating pest to tomato crops after its arrival in Europe in 2006 [11,12]. Since its detection in Europe, numerous researchers have determined the biology and the number of generations present or possible of *C. perspectalis* in those countries. Lopez and Eizaguirre [3] stated that the boxwood moth larvae mostly pupated after seven instars, although several larvae complete the larval stage after six instars. *C. perspectalis* overwinters as diapausing larvae in the third or fourth instar inside a silk cocoon over or between the leaves [3]. In Spain, *C. perspectalis* presents two complete generations and a partial third generation [3,13], as well as in other Mediterranean countries [14,15,16,17]. Fewer works have also been carried out to detect possible native biotic factors of mortality (predators, parasitoids, entomopathogenic microorganisms and nematodes) capable of attacking or infecting the pest in Europe; Oltean et al. [16] summarized data from the literature. The Diptera *Compsilura concinnata* (Meigen, 1824) was found to parasitize *C. perspectalis* larvae at a low rate in France [18]; Ferracini et al. [19] found another tachinid, *Pseudoperichaeta nigrolineata* (Walker, 1853), in Italy; and a sirphid species, *Xanthandrus comtus* (Harris, 1780), was found by Las Heras et al. [20] predating the larvae in Catalonia. Bird et al. [21] reported, for the first time in Britain, the parasitism of the boxwood moth by *P. nigrolineata* and by a pteromalid, *Stenomalina* cf. *communis* ((Nees, 18343). Zamani et al. [22] and Ghavamabad et al. [23] found larvae infected by *B. bassiana* in Iran. Finally, Peterlin, Rodic, and Trdan [24] mention predation by birds in Slovenia.

The effectiveness of biocontrol treatments has been tested in urban parks and gardens. Entomopathogenic microorganisms such as *B. bassiana* [25], *Bacillus thuringiensis* (Berliner) [20,26], viruses [27,28], and nematodes [20,29,30] have been studied. The insecticidal effects of some essential oils against the boxwood moth larvae [31] or oviposition-deterrent products, such as essential oils [32] or volatiles released from frass pellets of *C. perspectalis* [33,34], have also been tested, as well as the egg parasitoids of the genus *Trichogramma* [35]. Finally, the injection of insecticides into tree trunks, treatment with conventional systemic products [36] and leaf treatments [20,37,38] have also been studied.

Two very different approaches can be taken when considering possible control strategies of *C. perspectalis* in regions where *B. sempervirens* is an intrinsic element of natural forests but can also be found in urban and park gardens: in parks and gardens, treatments with indigenous natural enemies could be considered in more conventional applications (such as pulverizations or classical biological control). On the other hand, measures should be applied in forest areas to increase and conserve indigenous natural enemies that adapt to the pest.

This work aimed to collect and identify possible indigenous natural enemies in the boxwood of the southwest Mediterranean region and to test the efficiency of some of them for controlling *C. perspectalis* larvae in the first step under laboratory conditions.

## 2. Materials and Methods

### 2.1. Insect Rearing

For establishing a laboratory population of *C. perspectalis,* larvae were collected in May 2022 in two different areas of forest boxwood at Solsones (41°55′35.78″ N, 1°29′42.28″ E; and 41°57′06″ N, 1°31′42″ E). Larvae were fed on a semiartificial diet and kept at 25 °C, with a 16:8 (light: dark) photoperiod and 70% humidity until pupation. The pupae were separated by sex. When adults emerged, at least five pairs of male and female moths were put in each mating cage under the previous temperature and photoperiod with a 5 % sugar solution. Some *B. sempervirens* branches were added as laying substrates. Every day, sprigs were inspected for eggs, and leaves with eggs were removed daily and used for the different experiments.

### 2.2. Semiartificial Diet

The diet used by Lopez and Eizaguirre [3] produced a mortality percentage in the newly born larvae close to 50%. Many of the dead larvae did not taste the diet. To improve the palatability and larval survival, all of the components of the diet [3] (agar–agar, brewer’s yeast, wheat germ, freeze-dried leaves of *B. sempervirens*, ascorbic acid and sorbic acid) were tested in several preliminary experiments. Other standard diet components, such as benzoic acid, nipagin and aureomicin, were also tested. Finally, the diet based on lyophilized *B. sempervirens* leaves developed by Lopez and Eizaguirre [3] was simplified, as shown in Table 1. This diet produced a low mortality (close to 2%) of newborn larvae.

### 2.3. Detection of Natural Factors of Mortality

From 2018 until 2022, a total of 800 developed L5–L6 larvae were collected at different locations in Catalonia. Except for the 2022 sampling, which was conducted in May, the rest of the samplings were carried out in September (Table 2). Collected larvae were individualized, fed on a semi-synthetic diet based on *Buxus sempervirens* (Table 1) and kept in climatic chambers at 25 °C and a 16:8 (light:dark) photoperiod. All larvae were observed daily until they pupated or died. Emerged adult parasitoids were stored in 70% alcohol and sent to be identified by the specialists Hans-Peter Tschorsnig and Pierfilippo Cerretti. Dead larvae with symptoms of infection by entomopathogenic microorganisms were individualized and sent to the laboratory of Entomology of the Department of Agronomy of Cordoba University, where the possible control agents were identified and reared for later testing in the Lleida laboratory for their efficacy as *C. perspectalis* larvae control agent.

### 2.4. Fungal Strain and Inoculum Preparation

The *Beauveria bassiana* EABb 20/01-Cp strain held in the culture collection of the Department of Agronomy of Cordoba University, which was isolated from an infected larva of *C. perspectalis* from Ripolles (42°17′59.82″ N, 2°9′52.88″ E 2020 (Table 2), was used in this study. The microscopic identification was made using morphological characteristics and taxonomic keys [39,40]. The strain identification was molecularly confirmed with bases in the nuclear EF-1α gene. Genomic DNA was extracted using the method of Raeder and Broda [41]. An 1100 bp fragment spanning the 3′ 2/3 of the EF-1α gene was amplified with the following primers: tef1fw (5′-GTGAGCGTGGTATCACCA-3’) [42] and 1750-R (5′-GACGCATGTCACGGACGGC-3’) [43]. PCR amplification was performed in a total volume of 50 µL containing 15 µL of genomic DNA, 1X DreamTaqTM buffer with MgCl_2_ (ThermoScientific, Waltham, MA, USA), 20 nM of each primer, 40 nM dNTP mix and 0.25 U DreamTaqTM DNA polymerase (ThermoScientific, Waltham, MA, USA). The amplification conditions were as follows: initial denaturation at 94 °C for 1 min, followed by 35 amplification cycles (denaturation at 94 °C for 1 min 30 s, annealing at 55 °C for 2 min, extension at 72 °C for 3 min), followed by a final extension at 72 °C for 10 min. The PCR products were visualized on 1% agarose gel, purified using the Geneclean II kit system (QBiogene, Inc., Carlsbad, CA) and sequenced by Stab Vida (Caparica, Portugal).

For the phylogenetic analysis, the sequence was compared with the sequences of a further 22 *Beauveria* spp. strains and one *Cordyceps militaris* (L.) Fr. used as an outgroup. The sequence data from all 24 *Beauveria* strains were aligned using the MegAlign program (DNASTAR package, London, UK). The phylogenetic analysis was carried out using the MEGA X program [44,45]. The evolutionary history was inferred using the neighbor-joining method [46]. The percentage of replicate trees in which the associated taxa clustered together in the bootstrap test (1000 replicates) are shown next to the branches. The evolutionary distances were computed using the Tamura 3-parameter method and are in the units of the number of base substitutions per site [47]. All ambiguous positions were removed for each sequence pair (pairwise deletion option).

### 2.5. Efficacy of Beauveria bassiana in Controlling Cydalima Perspectalis Larvae

For bioassays, the *B. bassiana* EABb 20/01-Cp strain was grown for 15 days on malt agar in 90 cm diameter Petri plates at 25 °C in darkness. Conidial suspensions were prepared by scraping conidia from the agar into a sterile aqueous solution of 0.1% Tween 80, filtering through a piece of cheesecloth and vortex mixing to encourage conidia into suspension. Conidial viability was verified before the preparation of suspensions by germinating tests on malt agar supplied with 500 mg L^−1^ streptomycin sulfate salt (Sigma-Aldrich Chemie, China) after 24 h at 25 °C and always exceeded 90%. This procedure was also followed to obtain mycelia for DNA extraction but covering slants with a sterile cellophane sheet. The total DNA was extracted from lyophilized mycelia scraped from the cellophane sheets, according to [41].

#### 2.5.1. Bioassay with Sixth Instar (L6) Field-Collected Larvae

Approximately 250 larvae were collected from a population in the region of Solsones (41°55′35.87″ N 1°29′42.28″ E Catalonia) (Table 2). These were stored at 10 °C and with a 16:8 (light:dark) photoperiod until the time of treatment. Of the collected larvae, 180 of the L6 instar were selected for experiments. These field larvae were randomly divided into groups of ten larvae each. Of the 18 groups obtained, nine were treated, and the other nine groups were considered the control treatment, resulting in a total number of 90 larvae in each treatment. The treatment consisted of immersing the larvae of each group for 30 s in a 10^8^ conidia ml^–1^ fungal suspension (10 mL). The controls were similar to the treatments, but only immersing ten larvae of each group into a 10 mL sterile aqueous solution of 0.1% Tween 80 for 30 s.

Once bathed, *C. perspectalis* larvae were placed on filter paper for a few seconds to dry. Subsequently, they were placed individually in breeding boxes with an approximately 1.5 cm^3^ piece of the semiartificial diet. Larvae were kept at 20 °C, with a 16:8 (light:dark) photoperiod and 70% humidity until pupation and adult emergence. They were observed daily, the day of death, pupation or adult emergence was noted, and the sex and weight of the pupae were registered. The dead larvae and pupae were collected in Eppendorf tubes to be sent to the laboratory of Entomology of the Department of Agronomy of Cordoba University to confirm if the mortality factor was *B. bassiana* EABb 20/01-Cp strain. For that purpose, dead larvae were surface sterilized with 1% sodium hypochlorite followed by three rinses in sterile distilled water for 1 min each; then, they were placed on sterile wet filter paper in sterile Petri plates, sealed with laboratory film, incubated at 25 °C and inspected for fungal outgrowth. The experiment finished 25 days after treatment when all survival individuals emerged as adults.

#### 2.5.2. Bioassay with L2–L3 and L4–L5 Laboratory-Reared Larvae

Larvae from the laboratory rearing of different egg clusters were separated into groups of 10 larvae/cage and fed on the semiartificial diet from hatching until achieving the suitable age for treatments. The larvae were checked daily to detect the cephalic capsule that indicates the molt; when enough larvae of the cluster reached the L2–L3 or L4–L5 instar, the treatments were conducted in the same way and with the same conidia concentration as previously described. As in the previous experiment, for each larval age, nine groups of ten larvae each were treated with the control solution (aqueous solution of 0.1% Tween 80) and the other nine groups with the *B. bassiana* solution (total of 90 larvae for each larval age and treatment). Larvae were fed on the semiartificial diet and kept at 20 °C, with a 16:8 (light: dark) photoperiod and 70% humidity. After the treatment, they were observed daily for 25 days, and the day of death was noted.

### 2.6. Statistical Analysis

The Statgraphics Plus computer package [48] was used for the analyses. Analysis of the survival probability was performed for each larval age and treatment, and Logrank and Wilcoxon tests were calculated. Analysis of Variance (ANOVA) of the number of resulting live larvae in each repetition for each larval age and treatment was performed. The LSD test was used to separate means when needed. The influence of treatment on the number of larvae that produced a cocoon was analyzed with an ANCOVA, where the factors were the age of the larvae treated and the treatment, and the covariate was the number of larvae that survived the treatment.

## 3. Results

### 3.1. Presence of Parasitoids

The presence of parasitoids in the larvae collected for the different experiments from 2018 to 2021 is shown in Table 3. No parasitoids were found in 2018–2019 in Garrotxa, where the first attacks of *C. perspectalis* in Catalonia were detected. All parasitoids detected later belonged to *C. concinata* (Diptera: Tachinidae), a generalist parasitoid common in Lepidoptera larvae. The highest percentage of this species emerged from the larvae collected in Ripolles in 2020.

In 2022, in a sampling of wintering L6 larvae carried out in Solsona in May, 0.5% of the larvae also became parasitized by *C. concinnata*.

### 3.2. Identification of Beauveria bassiana Strain

A total of 0.5% of the dead larvae (Table 3) sent to the laboratory of Entomology of the Department of Agronomy of Cordoba University suffered an infection that was identified as being due to *B. bassiana*. Morphological features allowed the identification of the EABb 20/01-Cp strain in *Beauveria* genera. The mycelium was white with a fluffy appearance. The conidiophores were single and irregularly grouped with an apparent zigzag pattern after conidia production (rachis). Conidia were hyaline, rounded and single-celled. The sequence of 952pb was deposited in the Genbank database with accession number ON946215. The phylogenetic tree confirmed that the *Beauveria* EABb 20/01-Cp strain was aligned with the *B. bassiana* species (Figure 1). The sequence of the EABb 20/10-Cp strain was identical, aligning with the reference AY531893 and AY531962 strains of *B. bassiana*.

### 3.3. Efficacy of Beauveria bassiana in Controlling Cydalima Perspectalis Larvae

The survival probability of the larvae of the different instars treated with *B. bassiana* isolate is shown in Figure 2. In the three age groups considered, the percentage of surviving larvae treated with the control solution was higher than that of the larvae treated with the *B. bassiana* suspension (Table 4). The *B. bassiana* treatment produced a mortality percentage from 30% in the L4–L5 laboratory-reared larvae (Figure 2B) to near 100% in the field-collected L6 larvae (Figure 2C). The effect of the *B. bassiana* treatment was detectable from the third day on the L2–L3 larvae (Figure 2A), from the fourth day on the L4–L5 larvae (Figure 2B) and from the fifth day on the L6 larvae (Figure 2C). A total of 59% of the L6 larvae treated with *B. bassiana* solution survived until the pupal molt, but of these, 43% died as pupa due to the treatment applied.

The significant effects and the interactions of the number of alive larvae of different ages 25 days after treatment are shown in Figure 3. The age (*F* = 29.65, *p* < 0.001) and the treatment (*F* = 136.98, *p* < 0.001) affected the survival of the larvae. There was also an interaction between the age of the larvae and the treatment (*F* = 37.06, *p* < 0.001). The small differences in the number of alive larvae in the controls could be attributed to larval handling, but when the larvae were treated with *B. bassiana,* the number of live L6 larvae was much lower than the number of live L2–L3 or L4–L5 larvae, indicating the higher virulence of the treatment.

Several larvae from the bioassays with L2–L3 and L4–L5 larvae prepared a cocoon (Table 5), remaining inside it as diapausing larvae, ceasing feeding and development. None of the L6 instar larvae, neither treated nor untreated, prepared a cocoon. The ANCOVA showed that the treatment did not affect the number of L2–L3 or L4–L5 larvae that prepared a cocoon (*F* = 1.23, *p* = 0.28). However, there were significant differences in the number of cocoons prepared according to the age of the larvae that received the treatment (*F* = 11.98, *p* < 0.001) (Table 5).

## 4. Discussion

Two indigenous natural enemies, a parasitoid, *Compsilura concinnata*, and an entomopathogen, *Beauveria bassiana*, were detected in the samplings carried out in the natural boxwood forests in the Mediterranean region.

*Compsilura concinnata* (Diptera: Tachinidae), a common tachinid species in Europe [49], is a parasitoid found with some regularity in the collected larvae of *C. perspectalis* during the last three years. It has also been recently detected in France [50], Iran [51] and other regions of Spain (personal communication). Occasionally, another dipteran species, *Exorista larvarum* (L.), is reported in Romania as *C. perspectalis* [52], but Martini et al. [53] indicated that *C. perspectalis* is not a suitable host for this parasitoid. Although the most recent works on *C. concinnata* are related to the introduction of the parasitoid for the control of *Lymantria dispar* (L., 1778) in the United States, the species is native to Europe. It has been considered an important natural enemy for controlling forest pests such as *Thaumetopoea*
*processionea* (Denis and Schiffermüller, 1775) and *T. pytiocampa* (Denis and Schiffermüller, 1775) [54]; *L. dispar*, *Yponomeuta malinellus* (Zeller, 1838) or *Euproctis chrysorroea* (L., 1758) [55]. In boxwood forests, *C. concinnata* can be an excellent tool to stop the explosions in the number of the invader *C. perspectalis*. The species can develop three or four generations in different alternative hosts, including up to 200 species of the 17 families of Lepidoptera and some members of Hymenoptera or Coleoptera [56]. The species overwinters as a larva develops inside the host, which may be a pupa or a developed larva of the family Lepidoptera. Due to the small size of the overwintering larvae of *C. perspectalis*, the parasitoid cannot overwinter in this species. Therefore, its winter survival depends on the presence of suitable alternative hosts. On the other hand, controlling the populations of *C. perspectalis* in the Mediterranean boxwood forests through applying synthetic insecticides or bioproducts is unrealistic. Nevertheless, increasing natural parasitism due to *C. concinnata* by studying the presence of alternative hosts could be considered, mainly in winter or spring when *C. perspectalis* is not a suitable host for the parasitoid.

*Beauveria bassiana* (Ascomycota: Hypocreales) infections of *C. perspectalis* have been reported previously, which shows this fungal genus as an excellent candidate in the search for new mycoinsecticides for *C. perspectalis* control [22,57,58] in parks or gardens. The strain of *B. bassiana* obtained from the larvae collected in the forest was sequenced and aligned with the *B. bassiana* species and has shown a different virulence against *C. perspectalis* larvae depending on the age of the larvae. Although Bujanadze et al. [58] showed that the locally isolated *B. bassiana* strain produced 80% of mortality in L2–L5 larvae; it makes no distinction in the results between the larval stages. In our work, the different virulence levels against L2–L3 and L4–L5 larvae suggest that *C. perspectalis* early larval instars may escape fungal disease mainly due to a combination of two factors, the low instar duration (3–5 days in these larvae versus 8–10 days in L6 larvae) and the production of cocoons, indicators of larval diapause [3] and low levels of conidial germination and rapid ecdyses, which removed conidia before their germ tubes penetrated the host hemolymph [59].

As Lopez and Eizaguirre [3] pointed out, a high percentage of *C. perspectalis* larvae can enter diapause at moderately low temperatures even though the photoperiod to which they are subjected is a long day (16:8, light: dark) photoperiod. In the case of this work, the temperature at which the larvae developed, 20 °C, and the day-long photoperiod did not prevent several larvae from producing cocoons. Lopez and Eizaguirre [3] also noted that these diapausing larvae could molt several times within the cocoon; the possibility that the larvae in this trial molted and the production of cocoons may have influenced the lower effectiveness of the treatment with the L2–L3 and L4–L5 larvae.

Commercially available *B. bassiana* mycoinsecticides have a concentration in the range of 10^9^ to 10^11^ conidia ml^–1^ fungal suspension. Therefore, a fungal spray with a commercially available mycoinsecticide will always outperform our bioassay because of using 10- to 1000-fold higher dosages and also a formulation with optimum co-adjuvants guaranteeing the maximum inoculum per larva. The high virulence against *C. perspectalis* L6 larvae suggests that it may be a good candidate to apply when conventional ingestion products such as *Bacillus thuringiensis* or synthetic insecticides are ineffective. In addition, under natural conditions, the young larvae are usually protected in the cocoons they make or between the leaves, making them more inaccessible to treatments. The developed larvae are more exposed to contact treatments due to their size and method of feeding. Therefore, in boxwood of gardens and parks, an excellent time to treat developed larvae with *B. bassiana* may be after wintering, when the larvae develop to give rise to the first flight of adults. At this time, effective and respectful treatment of the environment, such as treatment with the natural strain of *B. bassiana*, could prevent the damage caused by the first generation of non-wintering larvae, which are usually the most damaging to boxwood.

## 5. Conclusions

Two indigenous natural enemies, a parasitoid, *Compsilura concinnata,* and an entomopathogen, *Beauveria bassiana*, were detected in the samplings carried out in the natural boxwood forests in the Mediterranean region. The *B. bassiana* strain was highly virulent against the developed larvae of *C. perspectalis*, which suggests that it may be an effective treatment in parks and gardens when the first overwintering *C. perspectalis* larvae are detected. On the other hand, improving the current knowledge about the parasitoid species’ ecology may help contain *C. perspectalis* populations in natural conditions.

## Figures and Tables

**Figure 1 insects-13-00781-f001:**
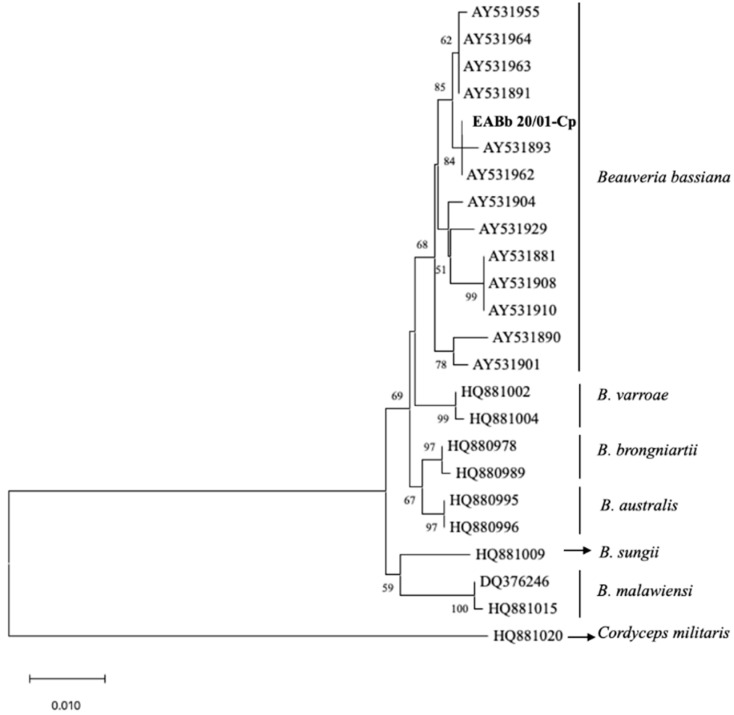
Phylogenetic tree of the EF1-α sequence from the Beauveria bassiana EABb 20/01-Cp (in bold) strain and 23 reference strains based on the neighbor-joining method using the Tamura 3-parameter method, with an outgroup and 1000 replicates of bootstrapping. A support threshold of 50% was set.

**Figure 2 insects-13-00781-f002:**
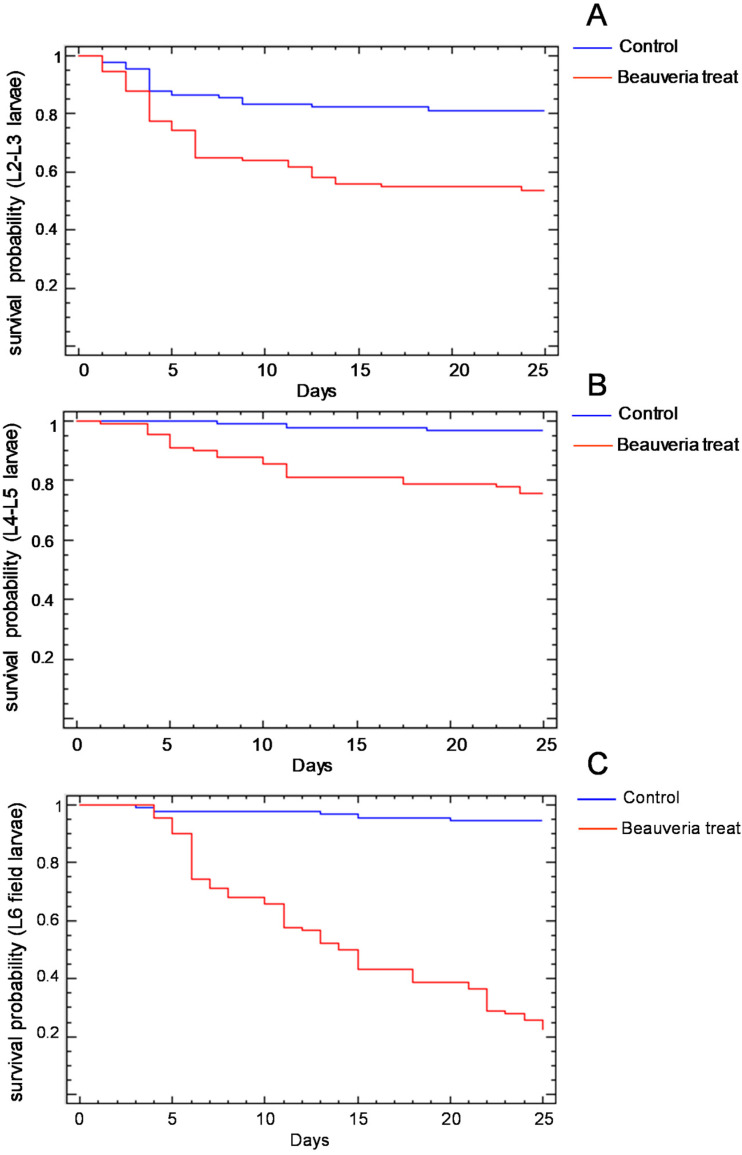
Survival probability of the L2–L3 (**A**), L4–L5 (**B**) and field-collected L6 (**C**) *Cydalima perspectalis* larvae treated with the identified strain of *Baeuveria bassiana* or with the control sterile aqueous solution.

**Figure 3 insects-13-00781-f003:**
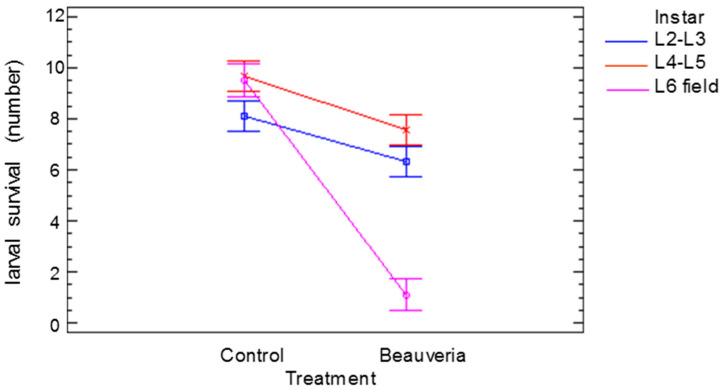
Significant effect (*p* < 0.05) of *Cydalima perspectalis* larval age (instar) treatment interaction on larval survival. Bars show the least significant differences.

**Table 1 insects-13-00781-t001:** Diet composition.

Components	% of Total Weight
Water	78.76
Agar–agar	1.8
Freeze-dried leaves of *Buxus sempervirens*	19
Benzoic acid	0.3
Nipagin	0.1
Aureomycin	0.04

**Table 2 insects-13-00781-t002:** Collected *Cydalima perspectalis* larvae in boxwoods of different regions of Catalonia and across different years.

Year	Area	Coordinates	Number of Larvae
2018	Garrotxa (Olot)	42°10′16.80″ N 2°28′51.84″ E	100
2019	Garrotxa (Olot)	42°10′21.36″ N 2°28′1.32″ E	150
2020	Ripolles (Ribes de Freser)	42°17′59.82″ N 2° 9′52.88″ E	150
2021	Ripolles (Fustanyà)	42°20′23.30″ N 2°10′35.60″ E	100
	Moianes (Sta Maria d’Oló)	41°52′4.76″ N 2° 1′33.72″ E	100
2022	Solsones (Solsona)	41°55′35.87″ N 1°29′42.28″ E	200

**Table 3 insects-13-00781-t003:** Percentage of parasitism and number of dead larvae due to unidentified agents of the collected larvae in the different regions of Catalonia and in different years.

Year	Area	% Parasitism	Number of Dead Larvae
2018	Garrotxa (Olot)	0	6
2019	Garrotxa (Olot)	0	8
2020	Ripolles (Ribes de Freser)	20	5
2021	Ripolles (Fustanyà)	0.5	12
	Moianes (Sta Maria d’Oló)	1	6
2022	Solsones (Solsona)	0.5	5

**Table 4 insects-13-00781-t004:** Statistics of the analysis comparing the survival probability of the larvae of different ages treated with the control solution with the survival probability of the larvae treated with the *Beauveria bassiana* solution.

Larval Instar	Logrank Test		Wilcoxon Test	
	χ^2^	*p*	χ^2^	*p*
L2–L3	14.87	<0.001	13.78	<0.001
L4–L5	16.93	<0.001	17.05	<0.001
L6 field	110.28	<0.001	97.67	<0.001

**Table 5 insects-13-00781-t005:** Mean number ± S.E. of the *Cydalima perspectalis* larvae that produced a cocoon, ceasing feeding and development, 25 days after the treatment. Number of repetitions is given in brackets.

	L2–L3	L4–L5
Number of larvae producing a cocoon	2.39 ± 0.28 (18) a	3.83 ± 0.28 (18) b
Total number of live larvae	7.22 ± 0.33 (18)	8.61 ± 0.33 (18)

Different letters indicate differences in the number of cocoons produced by the larvae of the different age groups.

## Data Availability

The data presented in this study are available in article.

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
