# Peer review of "Survey of Natural Enemies of the Invasive Boxwood Moth Cydalima perspectalis in Southwestern Mediterranean Europe and Biocontrol Potential of a Native Beauveria bassiana (Bals.-Criv.) Vuill. Strain"

_insects, 2022, doi:10.3390/insects13090781_

Round 1

Reviewer 1 Report (Previous Reviewer 1)

Compared to the previous version of the article, the new one looks better. However, there are still many shortcomings.

From Simple Summary and Abstract it is not clear what the essence of the article is and what the results of the study are. I also ask you to rewrite Conclusion. I have no idea how you can improve these key parts of the article.

The authors claim in 2.1: "To improve the palatability and the larval survival, the diet based on lyophilized Buxus leaves developed by Lopez & Eizaguirre [3] was simplified, as it is shown in Table 1". However, the table shows only the composition of the diet and does not explain what is changed and why these changes can improve the palatability and the larval survival. There are no explanations in the results either. In Results “The new diet used in these experiments (Table 1) has improved palatability and survival, producing mortality in newborn larvae lower than 98%. The mortality of newborn larvae feeding on the Box leaves was 96.8%”. Who and when conducted a comparative experiment on the survival of newborn larvae on different diets? There is no mention of these experiments in Materials and Methods. In addition, please explain why the mortality of newborn larvae is so high (96-98%). Perhaps it makes sense to describe this phenomenon in Introduction.

Line 39-41. The meaning of the last sentence is not clear, in addition, its expediency in this form in the abstract is doubtful.

Line 112. There is no dot at the end of the sentence.

Line 123. a least

Line 133. Table 1 an Table3

Lines 149-150. Replace 50 and 30 with 5' and 3' in 50-GTGAGCGTGGTATCACCA-30, 50-GACGCATGTCACGGACGGC-30.

Lines 270-272. How did you determine the starting day of the effect of the B. bassiana treatment? On what day in each age group were the differences between control and treatment statistically significant?

Line 344. Why is link 53 at the beginning of the paragraph? What follow are snippets of sentences.

Doi are not specified for some articles (for example, 20, 42, etc.).

Author Response

We want to thank the Reviewers for the thorough revision of the manuscript insects-1882260 and the helpful suggestions. We send a point by point answer to each one of the Reviewers Compared to the previous version of the article, the new one looks better. However, there are still many shortcomings.

ANSWER REVIEWER 1

From Simple Summary and Abstract it is not clear what the essence of the article is and what the results of the study are. I also ask you to rewrite Conclusion. I have no idea how you can improve these key parts of the article.

We have rewritten some sections. We hope that now it would be improved

The authors claim in 2.1: "To improve the palatability and the larval survival, the diet based on lyophilized Buxus leaves developed by Lopez & Eizaguirre [3] was simplified, as it is shown in Table 1". However, the table shows only the composition of the diet and does not explain what is changed and why these changes can improve the palatability and the larval survival. There are no explanations in the results either. In Results “The new diet used in these experiments (Table 1) has improved palatability and survival, producing mortality in newborn larvae lower than 98%. The mortality of newborn larvae feeding on the Box leaves was 96.8%”. Who and when conducted a comparative experiment on the survival of newborn larvae on different diets? There is no mention of these experiments in Materials and Methods. In addition, please explain why the mortality of newborn larvae is so high (96-98%). Perhaps it makes sense to describe this phenomenon in Introduction.

In fact, it was a mistake. According to the comment of the rest of the reviewers we have eliminated this paragraph but we have included more explanations in the section 2.2. Semiartificial diet.

We believe that is important to have a good diet to have a good rearing to do experiments in the laboratory.

Line 39-41. The meaning of the last sentence is not clear, in addition, its expediency in this form in the abstract is doubtful. It has been rewritten

This phrase has been rewritten

Line 112. There is no dot at the end of the sentence.

Added

Line 120. a least

Done

It is Table 1

Line 133. Table 1 an Table3

Lines 149-150. Replace 50 and 30 with 5' and 3' in 50-GTGAGCGTGGTATCACCA-30, 50-GACGCATGTCACGGACGGC-30.

Done

Lines 270-272. How did you determine the starting day of the effect of the B. bassiana treatment? On what day in each age group were the differences between control and treatment statistically significant?

We started the experiments when we have enough larvae of each age to do the treatments.

With the Life tables, Survival probability analysis it is difficult to determine when differences start to be

Line 344. Why is link 53 at the beginning of the paragraph? What follow are snippets of sentences.

Now this part of the manuscript has been changed.

Doi are not specified for some articles (for example, 20, 42, etc.).

Added

Reviewer 2 Report (Previous Reviewer 2)

Although the paper has improved, there is still a need to carefully look at several aspects. Some of my points are the same as in the first round of evaluation. I do not have time to point out all possible linguistic corrections, it would be good to have a specialist look at it. 

My detailed points:

L 63: …could later be adapted to the exotic species... Should be: …can later adapt to the exotic species

L 68-71. A natural response to the arrival of an invasive pest has been described with respect to Tuta absoluta (Lep. Gelechiidae), which turned out to be a devastating pest in tomato crops after it’s arrival in Europe in 2006:

-          Zappalá, L., Biondi, A., Alma, A., Al-Jboory, I.J., Arno, J., Bayram, A., Chailleux, A., El-Arnaouty, A., Gerling, D., Guenaoui, Y. & Shaltiel-Harpaz, L. 2013. Natural enemies of the South American moth, Tuta absoluta, in Europe, North Africa and Middle East, and their potential use in pest control strategies. J Pest Sci. 86: 635-647.

-          Crisol Martínez, E.; Van der Blom, J. (2019). Necremnus tutae is widespread and efficiently controls Tuta absoluta in tomato greenhouses in S-E Spain. IOBC-WPRS Bulletin, 147: 22-29.

L 77-79: In southwestern Mediterranean Europe, … as in other European countries such as Croatia [11], Hungary [12], Bulgaria [13], Italy [14], Spain [15], and Catalonia [16]. Spain belongs to SW Europe, and Catalonia is part of Spain. You can better say:

-          In Spain, C. perspectalis presents two complete generations and a partial third generation [3, 15, 16) as well as in other Mediterranean countries (11, 12, 13, 14).

L 108. As I remarked in my first comments: Adapt your M&M and results sections to the general course of events: Recollect larvae; isolate on artificial diet; rear out parasitoids or pathogens; set-up an insect rearing; conduct efficacy trials with an isolated pathogen. It is very confusing to see it in the reverse order, as you now present it. E.g., the first thing you describe is the new diet, whereas in the introduction this has not been presented as one of the objectives…  The order should be:

Detection of natural factors of mortality

Isolation and characterization of fungal entomopathogen

Bioassay: 1) inoculum preparation 2) bioassay with field collected larvae 3) Bioassay with lab reared larvae (here you describe the insect rearing).

Results:

L 228. You made a mistake… Instead of mortality lower than 98%, I suppose you mean mortality lower than 2% (or survival higher than 98%). Also: survival of new-born larvae feeding on the Box leaves…

L 265. I think Fig. 2 does not show the survival probability, but the survival. Also in Fig. 2.

I don’t understand what is actually reflected in Fig. 2. You have observed the larvae until they died, pupated or the adults emerged (L 196-197). In all cases (both with young larvae as with L6), your observations continue over a period of 25 days. Have all individuals hatched as adults or died by that time, or did you consider the ones that pupated before the end of the 25th day as final survivors? Please explain

L 270-271. Instead of The effect of the B. bassiana treatment was detectable …I suppose you should say: The differences in larval mortality became statistically significant…

L 272-274. I don’t understand this. In total, I see that approximately 80% of the treated larvae died. So, I suppose you want to say: …but of these 43% died as pupa. Don’t say here that it is due to the treatment, that is your conclusion. You can say that the difference in mortality between treated and untreated larvae is highly significant. The graph suggests that you could observe exactly when the pupae died…

Table 3 and L 287-289. P can never be 0.  Probably, you want to say P< 0,001. Please change.

L 299-300. Please correct: NONE of the L6 instar larvae, neither treated or untreated, prepared a cocoon.

Table 4. 1) What do you mean with ‘Total Nº alive larvae’? Does this include the ones that form a cocoon together with the ones that do not form a cocoon? 2) In the table, you don’t show possible differences between treated and untreated larvae. You can easily do that.

Discussion.

In your explanation towards the reviewer, you perfectly explain my questions about the formulation and efficacy of B. bassiana. Please mention the most important things in the paper (e.g. formulation of 10^9 in commercial products; efficacy of way of treatment).

L 353-355. First you mention that treatments with synthetic pesticides is unrealistic (L335), and immediately after (L 355), you suggest it as a possibility for application in winter (¿?).

Author Response

ANSWER REVIEWER 2

Although the paper has improved, there is still a need to carefully look at several aspects. Some of my points are the same as in the first round of evaluation. I do not have time to point out all possible linguistic corrections, it would be good to have a specialist look at it. 

My detailed points:

L 63: …could later be adapted to the exotic species... Should be: …can later adapt to the exotic species

Done, Lines 62-63

L 68-71. A natural response to the arrival of an invasive pest has been described with respect to Tuta absoluta (Lep. Gelechiidae), which turned out to be a devastating pest in tomato crops after it’s arrival in Europe in 2006:

-          Zappalá, L., Biondi, A., Alma, A., Al-Jboory, I.J., Arno, J., Bayram, A., Chailleux, A., El-Arnaouty, A., Gerling, D., Guenaoui, Y. & Shaltiel-Harpaz, L. 2013. Natural enemies of the South American moth, Tuta absoluta, in Europe, North Africa and Middle East, and their potential use in pest control strategies. J Pest Sci. 86: 635-647.

-          Crisol Martínez, E.; Van der Blom, J. (2019). Necremnus tutae is widespread and efficiently controls Tuta absoluta in tomato greenhouses in S-E Spain. IOBC-WPRS Bulletin, 147: 22-29.

Included, lines 71-74

L 77-79: In southwestern Mediterranean Europe, … as in other European countries such as Croatia [11], Hungary [12], Bulgaria [13], Italy [14], Spain [15], and Catalonia [16]. Spain belongs to SW Europe, and Catalonia is part of Spain. You can better say:

-          In Spain, C. perspectalis presents two complete generations and a partial third generation [3, 15, 16) as well as in other Mediterranean countries (11, 12, 13, 14).

Changed, lines 79-80

L 108. As I remarked in my first comments: Adapt your M&M and results sections to the general course of events: Recollect larvae; isolate on artificial diet; rear out parasitoids or pathogens; set-up an insect rearing; conduct efficacy trials with an isolated pathogen. It is very confusing to see it in the reverse order, as you now present it. E.g., the first thing you describe is the new diet, whereas in the introduction this has not been presented as one of the objectives…  The order should be:

Detection of natural factors of mortality

Isolation and characterization of fungal entomopathogen

Bioassay: 1) inoculum preparation 2) bioassay with field collected larvae 3) Bioassay with lab reared larvae (here you describe the insect rearing).

We have followed your suggestion in the order of the M&M section, results and discussion regarding to identification of mortality factors, and bioassays. However, when we start to work with a wild species it is essential to have a suitable diet and the rearing method designed when the larvae collection in the fields begins. For us, these are the first issues to solve and this is the reason we place them as the first point

Results:

L 228. You made a mistake… Instead of mortality lower than 98%I suppose you mean mortality lower than 2% (or survival higher than 98%). Also: survival of new-born larvae feeding on the Box leaves…

Yes, it was a mistake. According to the comment of the rest of the reviewers we have eliminated this paragraph but we have included more explanations in the section 2.2. Semiartificial diet.

Lines 124-133

We believe that is important to have a good diet to have a good rearing to do experiments in the laboratory.

L 265. I think Fig. 2 does not show the survival probability, but the survival. Also in Fig. 2.

Figure 2 shows the survival function, also called reliability function, that gives the probability that a study subject will survive beyond a given specific time. Normally, the survival function has to be estimated, and the standard way to do this is via Kaplan-Meier estimator. The Kaplan-Meier plot, has on the x-axis the time passed and on the y-axis the survival probability.

I don’t understand what is actually reflected in Fig. 2. You have observed the larvae until they died, pupated or the adults emerged (L 196-197). In all cases (both with young larvae as with L6), your observations continue over a period of 25 days. Have all individuals hatched as adults or died by that time, or did you consider the ones that pupated before the end of the 25th day as final survivors? Please explain

Several field collected larvae survived to the treatment and pupate. But after pupation we observed that some of them dead due to the treatment showing Beauveria infection symptoms (see the graphical abstracts). 25 days after the treatment all survival individuals (control or treated group) emerged as adults. So, we decided control the L2-L3 larvae for 25 days after treatment.

We have added the explanations in lines 218-219, 230-231

L 270-271. Instead of The effect of the B. bassiana treatment was detectable …I suppose you should say: The differences in larval mortality became statistically significant…

With survival probability function we don't know exactly when the differences start to be significant

I don’t understand this. In total, I see that approximately 80% of the treated larvae died. So, I suppose you want to say: …but of these 43% died as pupa. Don’t say here that it is due to the treatment, that is your conclusion. You can say that the difference in mortality between treated and untreated larvae is highly significant. The graph suggests that you could observe exactly when the pupae died…

Yes, corrected. Line 285

Table 3 and L 287-289. P can never be 0.  Probably, you want to say P< 0,001. Please change.

Changed in all places

L 299-300. Please correct: NONE of the L6 instar larvae, neither treated or untreated, prepared a cocoon.

Corrected, line 311

Table 4. 1) What do you mean with ‘Total Nº alive larvae’? Does this include the ones that form a cocoon together with the ones that do not form a cocoon? 2) In the table, you don’t show possible differences between treated and untreated larvae. You can easily do that.

Yes, The ANCOVA showed that there we not difference according with the treatment received, lines 311-313

Discussion.

In your explanation towards the reviewer, you perfectly explain my questions about the formulation and efficacy of B. bassiana. Please mention the most important things in the paper (e.g. formulation of 10^9 in commercial products; efficacy of way of treatment).

Added a paragraph in the discussion section, that has been rewritten lines 347-382

L 353-355. First you mention that treatments with synthetic pesticides is unrealistic (L335), and immediately after (L 355), you suggest it as a possibility for application in winter (¿?).

But these treatments are referred to parks and gardens. Clarification has been added in lines 349, 378

Rewritten, lines 369-375

Reviewer 3 Report (Previous Reviewer 3)

Dear Authors,

I found your manuscript improved but it still needs minor revision before acceptance for publication.

Please find my comments into your last pdf version (insects-1882260-peer-review-v1_second review_Aug2022.pdf) and the file: Second review of a Manuscript insects-insects-1882260_Aug 2022.docx.

Kind regards

Author Response

ANSWER REVIEWER 3

We want to thank the Reviewer 3 for the thorough revision of the manuscript and the helpful suggestions. We send a point by point answer to each one of the suggestions and a pdf with the correction suggested highlighted.

Second review of a Manuscript insects-1882260 with a title: Survey of natural enemies of the invasive boxwood moth Cydalima perspectalis in South-western Mediterranean Europe and biocontrol potential of a native Beauveria bassiana (Balsamo) Vuil. strain

by Carmen López , Sandra Las Heras , Inmaculada Garrido-Jurado , Enrique Quesada-Moraga , Matilde Eizaguirre 

Section: Other Arthropods and General Topics

Special Issue: Integrated Pest Management of Arthropods in Urban Green Spaces

Title: (Bals.-Criv.) Vuill.

Introduction:

Line 50: spreads

Done

Line 79: In the paper with reference No 13, the data are from Romania instead of Bulgaria.

Artola et al. (2018) reported three generations of Cydalima perspectalis (reference No 14)

Paragraph rewritten according with the Reviewer 2.

Line 79-87: Please specify if information about natural enemies of this species is for Europe.

Oltean et al. (2017) have summarized more data from the literature sources.

Included.

Line 85: The authors should give the names of the species parasitizing C. perspectalis larvae in Britain.

Done.

This should be more informative for the readers. Ferracini et al. (2022) (https://doi.org/10.3390/

f13020178) showed also the tachinid Pseudoperichaeta nigrolineata. Reference of this new publication to be added.

Included.

I recommend to the authors to read the paper of Wan et al (2014) (https://doi.org/10.1111/jen.12132). In the abstract, it is written that they have provided preliminary data on the parasitism of C. perspectalis in Europe.

We have read the paper but we consider that there are not substantialk information for this manuscript.

Line 92: I could not find information about essential oil in paper of Göttig and Herz (2017). It is about seasonal monitoring, sex dimorphism, and morphological variety. Please check.

Agree, eliminated.

Material and methods:

Line 130-131: By my opinion the locations and their coordinates with altitude should be presented here. Now these details are into Table 4., Results section.

We agree, we have added Table 2 in this section.

I still recommend to the authors to include the literature source they have used for identification of the parasitoid species in this part of the paper.

We sent the parasitoids to Hans-Peter Tschorsnig and Pierfilippo Cerretti, we have added this information in the manuscript.

Results:

Diet: Line 227-229: I still wonder if I misunderstand this paragraph:

“The new diet used in these experiments (Table 1) has improved palatability and 227 survival, producing mortality in newborn larvae lower than 98%. The mortality of newborn larvae feeding on the Box leaves was 96.8%. “

I don't think that a mortality of newborn larvae about 98% contribute to the survival of the larvae. This strange result should be discussed. This is very high mortality, even when you fed the larvae on Box leaves.

In fact, it was a mistake. According to the comment of the rest of the reviewers we have eliminated this paragraph but we have included more explanations in the section 2.2. Semiartificial diet.

We believe that is important to have a good diet to have a good rearing to do experiments in the laboratory.

3.4 Efficacy of Beauveria bassiana in controlling Cydalima perspectalis larvae

Line 287-289: Please specify if p < 0.001?

“(F = 29.65, p = 0.00) …. (F= 136.98, p = 0.00)  …(F = 37.06, p = 0.00)”.

The same for the line 303.

Done

Table 4: Add what does is mean the mark a and b on the first row with results.

Done

Conclusion after the second review:

I recommend the manuscript insects-1882260 to be accepted for publication in the scientific journal INSECTS after minor revision.

Comments are added into the file:

August 10, 2022

This manuscript is a resubmission of an earlier submission. The following is a list of the peer review reports and author responses from that submission.

Round 1

Reviewer 1 Report

The manuscript raises important and topical questions about natural enemies of the invasive boxwood moth Cydalima perspectalis and biocontrol using Beauveria bassiana. However, the paper has a number of serious shortcomings, which does not allow me to recommend this paper for publication.

1. The title, like the work as a whole, contains unrelated parts.

2. There is no study design to identify parasitoids and entomopathogens and justify the choice of collection sites for Cydalima perspectalis, so it is not surprising that there are very few results in this part of the manuscript.

In addition, I have a few questions and a lot of comments. Below I will give only a part of them as an example.

Introduction

Lines 71-72. Add references.

Line 80. What biocontrol treatments have been tested in urban parks and gardens and in which countries? Add references.

Materials and Methods

Line 169. How was conidial viability tested?

Line 183. Why 30 seconds but not 60 seconds as with conidia treatment?

Figures

Fig. 1 - It is necessary to highlight the strain with which the authors worked.

Fig. 3. If there were 10 larvae in each group, it is not clear why there are more than 10 specimens in the figure in the L6 field control.

Tables

In tables 2 and 3, fractional values are not rounded in an orderly manner.

Lines 110, 252. Why this information?

Citations in the text, as well as the list of references, are not designed according to the rules.

Punctuation marks are in disarray, a lot of missing and redundant ones.

Typos: line 246 de, line 251 bassina.

Lines 140-142: 50-…-30

Line 35: caterpillars. Larvae?

Line 131. C. larvae

Line 133. A Latin name at the beginning of a sentence is not abbreviated.

The meaning of some sentences is not clear, for example:

Lines 33-34: The samples allowed the detection and recollection of larvae infected with an entomopathogen

Lines 139-140: An 1100 bp fragment spanning the 30 2/3 of the EF-1a gene

Lines 214-216: The new diet used in these experiments (Table 1) has improved palatability and survival as it has produced a mortality in the new born larvae lover than 98% similar to the mortality of larvae feeding on the Box leaves. 

Reviewer 2 Report

The paper contains interesting information on natural control mechanisms of an invasive pest. The data show that, in spite of the recent arrival of the pests, several native organisms have responded to this newly present species.

The structure of the report should be improved a lot. Although there was a logical order in the work that was carried out, surveys in the first years; identification and lab rearing in the second phase and bioassays in the last phase. This can be better reflected in the text and better specified. For the ‘Methods’, I would suggest:

1-      In 2018 and 2019, a total of … larvae was collected at … different locations of Catalunia (¿different periods of the year?). These larvae were taken to the laboratory, where they were isolated in climatic conditions, at 25ºC and a 16: 8 (Light: Dark) photoperiod and fed on a semi-synthetic diet based on Buxus sempervirens (Table 1). All larvae were observed daily until they pupated or died. Emerged adult parasitoids were stored in 70% alcohol (¿?) and sent to be identified by specialist researchers. Dead larvae with symptoms of infection by entomopathogenic microorganisms were individualized and sent to the laboratory of Entomology of the Department of Agronomy of Cordoba University, where the possible control agents were identified.

2-     Identification of the pathogen. The detected pathogens were identified, both through morphological analysis as through gene sequencing. (Etc.)

3- Insect rearing and Bioassays. Could you indicate why this concentration (10^8) of B. bassiana was chosen and the time of treatment (60 seconds submerged)?      

Results. 

1-      Start with current table 5, adding a column for the Nº of larvae that died due to a pathogen.

2-      From … out of the … larvae that showed symptoms of death through an entomopathogen, the pathogen was identified as B. bassiana. Morphological features showed that etc. The detected pathogen was sequenced. The sequence of 952pb, characterized as strain EABb 20/01-Cp, was deposited in the Genebank database with accession number ON946215. The phylogenetic tree confirmed that the Beauveria EABb 20/01-Cp strain was aligned with the B. bassiana species (Figure 225 1). The sequence of EABb 20/10-Cp strain was identical to the strains with reference AY531893 226 and AY531962 of B. bassiana.

3- Presence of parasitoids...

4- Bioassays. The tables with the statistical details  (table 2 and 4) do not provide any extra information. These can be taken out. If you simply mention the kind of analysis that was carried out and the level of significance (N.S.; P< ...), it is enough. 

5. Table 3. With very simple statistics, G-test of independence, you can show that indeed the Nº of larvae that made a cocoon was not different between the treated larvae and the control. NS in all three cases. In this table, I see that the total Nº of L2-L3 in the control group must have been smaller, n=70, than in the other groups (n=90). This should be mentioned. Small mistake: Any L6 larvae produced a cocoon... (L. 257) should be No L6 larvae produced a cocoon... 

6. The data of figure 3 should be discussed before you start about the cocoons, because it is based on the same data as Fig. 2. Fig. 3 can be skipped, if you simply indicate the significant differences in Figure 2 (a, b, c...)

Discussion.

I miss some comments about the efficacy of the B. bassiana treatments. The treatment, 60 seconds submerged, is probably much more efficient than what you would get with a foliar treatment in the field. Do you have any reference from other trials with B. bassiana to support that this treatment may be comparable to a treatment in the field? Is 10^8 / ml also a concentration of commercial strains of B. bassiana when used against other pests? If not, you could say that the present results provide a first indication that B. bassiana may be a potential control agent, but that field trials with foliar applications are necessary. 

Reviewer 3 Report

Review of a Manuscript insects-1833329 with a title: Survey of natural enemies of the invasive boxwood moth Cydalima perspectalis in South-western Mediterranean Europe and biocontrol potential of a native Beauveria bassiana (Balsamo) Vuil. strain

by Carmen López , Sandra Las Heras , Inmaculada Garrido-Jurado , Enrique Quesada-Moraga , Matilde Eizaguirre 

Section: Other Arthropods and General Topics

Special Issue: Integrated Pest Management of Arthropods in Urban Green Spaces

The authors reported new data about natural enemies of the box tree moth, Cydalima perspectalis, which is invasive species in Europe causing damages on Buxus spp. in parks, gardens, and forests. One entomopathogenic fungus (Beauveria bassiana) and one parasitoid species (Compsilura concinnata) were found during a survey of natural enemies of this pest in southwestern Europe in the period of 2018 – 2022. The B. bassiana strain was sequenced. In addition to this, bioassays were performed under laboratory condition to test the effectiveness of the isolated strain of the entomopathogenic fungus on larvae of different ages – laboratory reared L2-L3 and L4-L5 larvae and field-collected L6 larvae. The results of the laboratory experiments showed the significantly higher survival of larvae in the control treatment in comparison with the fungal treatment at a concentration of 108 conidia/ml, and that L6 larval instar was most susceptible larval stage. The results obtained are a base for future researches for effective application of B. bassiana for control of C. perspectalis.

Methods used in this study are appropriate. Although the manuscript is well written I found many parts to be edited and improved. My comments are numerous and inserted into the attached pdf file with a name:

Some of the main remarks are presented bellow:

Title:

I recommend a modified title: “Survey of natural enemies of the invasive boxwood moth Cydalima perspectalis in southwestern Mediterranean Europe and biocontrol potential of a native Beauveria bassiana (Bals.-Criv.) Vuill. Strain”

Abstract:

I suggest corrections – see the pdf file. For your own results use past tense.

Introduction:

As authors have performed bioassays with different larval instars of C. perspectalis short information about life cycle of this species should be included.

I found some publications relevant to this study, which have been missed in the Introduction part. For example:

Bird S., Raper C., Dale-Skey N., Salisbury, A. 2020. First records of two natural enemies of box tree moth, Cydalima perspectalis (Lepidoptera: Crambidae), in Britain. British Journal of Entomology and Natural History 33: 67-70.

Ghavamabad, Reihaneh Gholami, Talebi, Ali Asghar, Mehrabadi, Mohammad, Farashiani, Mohammad Ebrahim and Pedram, Majid. "First record of Oscheius myriophilus (Poinar, 1986) (Rhabditida: Rhabditidae) from Iran; and its efficacy against two economic forest trees pests, Cydalima perspectalis (Walker, 1859) (Lepidoptera: Crambidae) and Hyphantria cunea (Drury, 1773) (Lepidoptera: Erebidae) in laboratory condition" Journal of Nematology, 53, 1, 2021, pp. 1-16. https://doi.org/10.21307/jofnem-2021-035

Gokturk, T., Chachkhiani-Anasashvili, N., Kordali, S. et al. Insecticidal effects of some essential oils against box tree moth (Cydalima perspectalis Walker (Lepidoptera: Crambidae)). Int J Trop Insect Sci 41, 313–322 (2021). https://doi.org/10.1007/s42690-020-00209-5

Miladinović, Z.; Mitrić, S.; Jakšić, B.; Nježić, B. Evaluation of potential of four entomopathogenic nematodes to control box tree moth (Cydalima perspectalis Walker). Book of Proceedings, XI International Symposium on Agricultural Sciences AgroReS 2022, pp. 158-165.

Szelényi, M.O.; Erdei, A.L.; Jósvai, J.K.; Radványi, D.; Sümegi, B.; Vétek, G.; Molnár, B.P.; Kárpáti, Z. Essential Oil Headspace Volatiles Prevent Invasive Box Tree Moth (Cydalima perspectalis) Oviposition—Insights from Electrophysiology and Behaviour. Insects 2020, 11, 465. https://doi.org/10.3390/insects11080465

Material and Methods:

Diet: Please check the total percent of the ingradients of the diet.

Detection of natural factors of mortality: Literature source about parasitoid identification should be added.

I suggest changes it the subtitle of 2.5.1 and 2.5.2. Some details in these parts are needed.

Results:

Diet: I found discrepancy in this interpretation:

The new diet used in these experiments (Table 1) has improved palatability and 214 survival as it has produced a mortality in the new born larvae lover than 98% similar to 215 the mortality of larvae feeding on the Box leaves.”

3.3 Efficacy of Beauveria bassiana in controlling Cydalima perspectalis larvae

The scale of x-axes of Figure 2A and Figure 2B should be the same as in Figure 2C. Please see the tick marks

Table 3: I found discrepancy between the table and the relative text. Please check.

Figure 3: to be removed. Results from ANOVA analysis is shown in Table 4.

3.4 Presence of parasitoids. This part need to be rewritten. Please see my suggestion.

Discussion:

It is worth to be mentioned that Compsilura concinnata is a common tachinid species in Europe (see Tachi et al. 2021 https://doi.org/10.1016/j.aspen.2021.01.001

References:

About some discrepancies in the literature sources, please see the attached file.

General comments:

The author’s names of the species mentioned in the manuscript should be added.

In the whole manuscript the authors use the two combinations: natural native (three times) and native natural enemies (six times).

I prefer the use of indigenous natural enemies instead of these combinations.

Many dots have appeared in the text, probably instead of commas.

The English needs improvements. Some suggestions are into the attached file.

I recommend the manuscript insects-1833329 to be accepted for publication in the scientific journal INSECTS after taking into account my remarks. Although they are numerous, I consider the suggested changes as minor ones.
